# Vibrational Spectroscopy as a Tool for Bioanalytical and Biomonitoring Studies

**DOI:** 10.3390/ijms24086947

**Published:** 2023-04-08

**Authors:** Sergey K. Pirutin, Shunchao Jia, Alexander I. Yusipovich, Mikhail A. Shank, Evgeniia Yu. Parshina, Andrey B. Rubin

**Affiliations:** 1Faculty of Biology, Shenzhen MSU-BIT University, No. 1, International University Park Road, Dayun New Town, Longgang District, Shenzhen 518172, China; pirutin@yandex.ru (S.K.P.); jsc@smbu.edu.cn (S.J.); rubin@biophys.msu.ru (A.B.R.); 2Faculty of Biology, Lomonosov Moscow State University, GSP-1, Leninskie Gory, 119991 Moscow, Russia; yusipovich@biophys.msu.ru (A.I.Y.); parshinae@gmail.com (E.Y.P.); 3Institute of Theoretical and Experimental Biophysics of Russian Academy of Sciences, Institutskaya St. 3, 142290 Pushchino, Russia

**Keywords:** infrared spectroscopy, Raman spectroscopy, vibrational spectroscopy in environmental studies, biological methods of environmental measurements

## Abstract

The review briefly describes various types of infrared (IR) and Raman spectroscopy methods. At the beginning of the review, the basic concepts of biological methods of environmental monitoring, namely bioanalytical and biomonitoring methods, are briefly considered. The main part of the review describes the basic principles and concepts of vibration spectroscopy and microspectrophotometry, in particular IR spectroscopy, mid- and near-IR spectroscopy, IR microspectroscopy, Raman spectroscopy, resonance Raman spectroscopy, Surface-enhanced Raman spectroscopy, and Raman microscopy. Examples of the use of various methods of vibration spectroscopy for the study of biological samples, especially in the context of environmental monitoring, are given. Based on the described results, the authors conclude that the near-IR spectroscopy-based methods are the most convenient for environmental studies, and the relevance of the use of IR and Raman spectroscopy in environmental monitoring will increase with time.

## 1. Introduction

Various types of vibrational spectroscopy (generally, it includes various variants of Raman and infrared spectroscopy) have been used for a long time to evaluate a variety of biological objects (see, for example, reviews [1,2,3,4,5,6]). Moreover, using vibrational spectroscopy, it is possible to evaluate individual compounds, cells, tissues, multicellular organisms (both living and fixed), and the products of their vital activity. These techniques are used for the assessment of the qualitative and quantitative composition of substances in studied biological objects and the conformations of compounds composing them. Among the advantages of these methods, one can mention their relative non-invasiveness, their significant experience in the subsequent analysis of results, and the possibility to perform in situ and in vivo measurements. The cost of sample preparation and the price of the equipment vary in a very wide range, from fairly cheap to very expensive. The possibilities provided by this group of research methods are in demand in various areas of biology, medicine, and the food industry.

Currently, vibrational spectroscopy is not widely used for biological environmental studies, although many of them are excellent for this. To realize such an approach, it is necessary, if possible, to perform a noninvasive evaluation of various characteristics of living organisms and their products, as well as to rapidly treat the obtained information; the cost of the analysis should be as low as possible [7]. Various types of vibrational spectroscopy meet the above-described requirements.

This review briefly describes the possibilities of Raman spectroscopy (RS) and infrared (IR) spectroscopy as research methods as well as the results of a number of studies, which can be used and/or are already used as tools for biological studies and environmental monitoring.

## 2. Biological Methods of Environmental Monitoring

Environmental monitoring is a very broad multidisciplinary concept that includes the monitoring of the state of the environment and its components and occurring processes using various methods of assessment and forecasting of changes [8]. Among the methods used for environmental monitoring, one can especially mention biological methods intended to evaluate changes in various biological objects and/or their components in response to the state of the environment or its changes.

Biological methods of environmental monitoring can be conditionally divided into (*a*) bioanalytical (laboratory) methods, i.e., the use of biological substances for environmental analyses (biosensors, bioassays, and biotests), and (*b*) biomonitoring (bioindication), i.e., the evaluation of abiotic, biotic, and anthropogenic factors using biomarkers, bioindicators, and biomonitors (Figure 1) [9].

Analyzing the existing methods and approaches, one can separately mention such a biomethod as human biomonitoring, which is actively developed now as an independent area of ecological monitoring [10,11,12,13]. This is a quite specific area belonging to more medicinal than ecological tasks, so it is not included in this review. Such methods were described in detail in several reviews [14,15,16].

### 2.1. Bioanalytical (Laboratory) Methods

Biosensors are devices that contain biomaterials, or materials of biological origin, that are used to identify various chemical compounds. Biotesting under laboratory conditions can be performed by various methods, but all of them can be described by three main approaches [17]: (*a*) toxicity tests, where a pollutant is injected into a clean environment containing biological samples; (*b*) comparison of the toxicity of real samples (taken from the environment) with that of standard samples, i.e., toxicity tests based on real samples; and (*c*) in situ testing using various organisms placed in the studied environment.

Bioassays (biotests) are based on the use of living test species, which are directly affected by the environment (such as the soil, surface water, etc.) to measure a potential biological effect of the presence of potential pollutants [18]. A number of bioassays intended for toxicity assessment have been standardized and are commercially available; various microorganisms can be used in bioassays, making it possible to evaluate such parameters as population growth, substrate consumption, respiration, adenosine triphosphate (ATP) luminescence, bioluminescence inhibition, genotoxicity, bioaccumulation of chemicals from the environment or samples, etc. [19].

### 2.2. Biomonitoring

Biomonitoring methods are usually used in addition to the analytical methods of environmental control based on various physical and chemical parameters. Today, the system of biomonitoring of the aquatic environment is the most developed [20,21,22].

The main feature of the indicator species is their ability to survive in a narrow range of actions from any external factor (the so-called tolerance zone), thus indicating the presence of this factor in the environment via the presence or absence of the species itself [23]. If any external factor provides a negative influence on the bioindicator species, then it is considered a stress factor. Stress represents a bioindicating organism’s response to non-optimal environmental parameters, which may have varying durations or intensities. The determination and ascertainment of this response type in a bioindicator represent the main goals of the bioindication methods [24]. Bioindicator response may be manifested via changes in population size or in the morphology and physiology of bioindicating organisms, such as changes in the content of pigments and lipid droplets, starch accumulation, etc. [24].

Bioindicator species may include various organisms, such as plants, aquatic species, and certain fish populations [24]. Plant organisms (e.g., mosses and algae) as well as fungi and lichens are used to evaluate atmospheric sediments, soil quality, and water purity [25]. Aquatic organisms and fish populations can also provide information about water quality [26].

Bioindicators are divided into the following types: (*a*) pollutant bioindicators (plants, animals, fungi, etc.); (*b*) ecological bioindicators used to evaluate the fragmentation of their habitat or other stresses (lichens, microalgae, plants, fungi, etc.); (*c*) biodiversity indicators (plants, animals, fungi, microorganisms, etc.) used to assess the species richness, endemism, genetic parameters, population-specific parameters, and landscape parameters; and (*d*) environmental bioindicators (invertebrates, marine and coastal species, etc.) used to evaluate or monitor changes in the environment [24].

One of the main advantages of the use of bioindicators is the possibility of observing the biological effects of various pathogenic factors and pollutants in a wide range of doses, as well as the temporal integration of contaminating conditions at the site of sampling and information collection.

According to the World Health Organization, bioindicators should be distinguished from biomonitors [9]: bioindicators reveal the presence or absence of a pollutant by the appearance of typical symptoms or measurable reactions, while biomonitors provide not only information about the presence of a pollutant but also additional information about the volume and intensity of exposure.

In addition, biomonitoring uses such tools as markers [27]. Biomarkers represent any measurable traits of a marker organism (any morphological changes, an increase or decrease in the concentration of a substance, etc.) responding to the presence of a toxicant.

The responses of biosensors, bioassays, and biotests as well as bioindicators can be tracked in various ways. There can be laboratory and field studies characterized by different approaches. In the case of field studies, the priority is given to the simplicity of measurements and obtaining results, the rapid procedure accomplishment, and the inexpensiveness of both equipment and sample preparation procedures. The obtained result should be easily interpreted (“yes” or “no”, “infected” or “not infected”, “belong to one of these groups”, etc.). Not only the presence of results but also the measurement of morphological characteristics is important. However, in the case of field studies, the amount of the obtained information and its accuracy and reliability are lower than those obtained under laboratory conditions.

For the majority of research methods used in environmental monitoring, the obtaining of a sufficient amount of samples is very important, though it is not always possible. In this aspect, approaches based on vibrational spectroscopy can be useful since they do not require a large amount of the studied material.

## 3. Vibrational Spectroscopy and Its Types: A Brief Theory

### 3.1. Basic Principles

#### 3.1.1. Wavelength and Frequency

There is electromagnetic radiation that can be considered a wave or a particle moving at the speed of light and having a certain wavelength and frequency. The frequency ν is the number of wave cycles passing through a point per second and is measured in Hz (cycles/second); the wavelength λ is the length of a full wave cycle, which is often measured in spectrometry in cm or nm. The wavelength and the frequency are related by the following ratio [28]:ν = c/λ,(1)
where c is the speed of light (3 × 10^10^ cm/s).

#### 3.1.2. Energy Quantization: Wave Number

The concept of corpuscular-wave dualism assumes that the energy of electromagnetic radiation is a quantized value, i.e., takes discrete values. The energy value, E, of one quantum can be calculated by the following formula:E = h · ν = h · c/λ,(2)
where h is the Planck constant (6.6 × 10^−34^ J·s).

Another parameter used in the spectroscopy is a wave number, ν, equal to 1/λ and corresponding to the number of waves per 1 cm. This parameter is measured in cm^−1^ and sometimes it is called frequency.

#### 3.1.3. Selection Principle

The chemical bonds in a molecule are subject to various kinds of fluctuations—changes in the length or the angles between bonds. These vibrations contribute to the total energy of the molecule. Since the energy of a molecule is quantized, the molecule is characterized by a certain set of energy sublevels, which can be calculated according to the rules of quantum mechanics using the equation [28]:E = (n + 1/2) · h · ν,(3)
where n is a quantum number (0, 1, 2, 3, …) corresponding to different energy levels. There is a so-called selection principle that determines the possibility of transitions between different energy levels. According to this principle, only transitions to the next energy level are allowed; therefore, molecules will absorb (give up) an amount of energy equal to hν. This rule is not strict, and sometimes transitions equal to 2hν, 3hν, or higher are observed. They correspond to spectral bands called overtones (Figure 2a) and are characterized by a lower intensity than the main vibration bands.

#### 3.1.4. Types of Molecular Spectra

The energy of a molecule can be represented as the sum of energies of individual types of movements, such as the electron movement around the nuclei (E_el_), vibrations of nuclei around their equilibrium positions (E_vibr_), and rotation of a whole molecule in the space (E_rot_) [30]:E = E_el_ + E_vibr_ + E_rot_,(4)

Note that the E_el_ value exceeds the E_vibr_ value by several orders, and E_vibr_, in turn, is several orders of magnitude higher than E_rot_ (E_el_ ≫ E_vibr_ ≫ E_rot_).

The spectra of a molecule represent the total energy (ΔE) absorbed, released, or scattered during numerous transitions of atoms or molecules from one energy state to another quanta. In this case, the total frequency of absorbed or emitted light is calculated by the following formula:(5)$$\overset{-}{\nu }=\Delta E/h\cdot c=E_{\mathrm{el}}/h\cdot c+E_{\mathrm{vibr}}/h\cdot c+E_{\mathrm{rot}}/h\cdot c,$$

The electron-vibrational-rotational (also called electronic) molecular spectra are registered in the ultraviolet and visible regions (see Table 1) [30]. Individual bands of these spectra correspond to different ΔE_vibr_ and/or ΔE_rot_ for a given ΔE_el_, and/or ΔE_el_ and ΔE_vibr_, respectively.

If the electronic state of a molecule does not change when interacting with radiation (ΔE_el_ = 0), it generates a vibrational-rotational (vibrational) spectrum. The term “vibrational spectrum” is applied to the spectra obtained by infrared and Raman spectroscopy.

#### 3.1.5. Characteristic Frequencies

An empirical approach based on group (characteristic) frequencies, commonly known as such, is used to interpret vibrational spectra. It was experimentally established that the presence of certain functional groups in a molecule causes the appearance of frequencies with a certain wavelength (wavenumber), or characteristic frequencies. Vibrations of the other parts of the same molecule insignificantly (~5%) affect the magnitude and position of these characteristic frequencies [30]. In the case of violation of the selection principle, bands with a higher wavenumber and lower intensity (overtones) may arise (see above). In addition, during vibrational transitions from excited states, molecules may generate so-called combinational (composite) bands with a frequency equal to ν_1_ + ν_2_, where ν_1_ and ν_2_ are group frequencies of any functional group.

#### 3.1.6. Classification of Molecular Vibrations

Molecular vibrations can be divided into stretching vibrations, which change the length of bonds, and bending vibrations, which result in changes of angles between the bonds [29]. For small molecules or groups of atoms in the composition of large molecules, stretching vibrations can be divided into symmetric (simplified, vibrations of bonds increase or decrease simultaneously) and antisymmetric (some bonds are stretched while other bonds in a group of atoms are shortened). Bending vibrations (especially if one part of the molecule is larger than the other one) are divided into in-plane bending and out-of-plane bending; in-plane bending vibrations, in turn, are divided into scissoring and rocking, and out-of-plane ones are divided into wagging and twisting. Vibrations, which consist in the simultaneous change of the lengths or valence angles of several bonds, are called skeletal.

### 3.2. Vibrational Spectra

Vibrational spectroscopy is the molecular spectroscopy branch, which studies the absorption and reflection spectra provided by quantum transitions between the vibrational energy levels of molecules. The frequency range of vibrational transitions expressed in wave numbers ranges from 100 to ~40,000 cm^−1^ (near the UV, visible, and IR ranges).

#### 3.2.1. Origin of Spectra

Vibrational spectra of compounds arise either as a result of a direct absorption of infrared radiation converted then to change the vibrational state of molecules (IR spectroscopy) or during an inelastic interaction of visible and/or ultraviolet radiation with a molecule, when a part of the light energy is absorbed or emitted by the molecule that leads to a change in the vibrational state of molecules (Raman spectroscopy) [4].

#### 3.2.2. Intensity of Bands

During this process, the intensity of the frequency band of the Raman spectra is determined by the magnitude of changes in the molecular polarizability (the ability of compounds to acquire an electric or magnetic dipole moment in an external electromagnetic field) when this molecule vibrates at this frequency. The intensity of a line in the IR spectrum depends on the ability of molecular vibrations to be excited by light of the corresponding frequency as well as on the ability of radiation to change the dipole moment of the molecule. The ability to change the dipole moment and polarizability can be differently expressed for different functional groups, and, therefore, their characteristic bands in the IR and Raman spectra will be characterized by different intensities [3].

#### 3.2.3. Using Spectra

Using vibrational spectra, one can estimate the presence and conformation of certain groups in a molecule. In addition, vibrational spectra can be used for the evaluation of crystal structures, whose presence and structure affect the energy of vibrational transitions and, therefore, the vibrational spectrum. The spectrum intensity allows one to judge the number of molecules of a given type in a sample and/or the number of bonds of a certain type.

### 3.3. Infrared Spectroscopy

Infrared (IR) spectroscopy is a type of absorption spectroscopy that is used to study the interaction of IR radiation with various substances. The result of the measurement performed by various types of IR spectroscopy is the IR absorption spectrum, i.e., the dependence of the intensity of the transmitted IR radiation (transmission, T) or optical density (D) on its frequency (wave number) [31]. Under certain conditions, the radiation absorption by a compound in the IR region can be quantitatively described by the Booger-Lambert-Behr law.

#### 3.3.1. Mid-, Near-, and Far-IR Spectroscopy

Based on the studied range, IR spectroscopy can be divided into three regions: near-IR, mid-IR, which is sometimes called IR spectroscopy, and far-IR (see Table 1) [29]. Low-energy long-wave radiation within this range provides enough energy to cause transitions between vibrational sublevels (Figure 2a). As a rule, these are transitions from the main vibrational sublevel to the first fundamental transition [28].

Far-IR spectroscopy is rarely used for structural studies, especially in the case of biological objects, but is useful for obtaining information about the vibrations of structures containing atoms of “heavy” elements (e.g., Fe–O, S–S), skeletal vibrations (which involve several intramolecular bonds), and crystal lattice vibrations [29].

On the contrary, various types of near- and mid-IR spectroscopy are actively used for the study of biological samples. In the case of the near-IR region, the bands corresponding to the overtones of characteristic frequencies or the composite bands are usually observed [4]. The mid-IR region is divided into the group frequency region (4000–1500 cm^−1^) and the fingerprint region (below 1500–600 cm^−1^). The name of the fingerprint region is associated with the fact that the position and intensity of absorption bands in this region are strictly individual for each organic compound. Vibrations common to various functional groups are manifested in the first region: vibrations involving hydrogen atoms (from 4000 to 2850 cm^−1^), vibrations of various triple bonds (from 2500 to 2000 cm^−1^), vibrations of various double bonds (from 2000 to 1500 cm^−1^), etc. [29]. In the fingerprint region, various types of vibrations typical for C–C, C–N, and C–O bonds are observed, as are bending vibrations for X–H: C–H, O–H, and N–H bonds, including bending and skeletal vibrations of polyatomic systems.

The bands of the mid-IR spectrum are associated mainly with vibrations of various functional groups of molecules, while bands of the near-IR spectrum are associated with overtones and combination bands. Compared to mid-IR spectroscopy, the band intensity of near-IR spectra is significantly (several orders of magnitude) lower (low sensitivity), and the spectra have much more bands, often overlapping each other, which significantly complicates their treatment, separation of bands, and interpretation. However, unlike the mid-IR spectroscopy, sample preparation for the near-IR spectroscopy is rather minimal, and the spectra are almost not affected by the presence of water, atmospheric CO_2_ (water, as a dipole molecule, actively absorbs IR radiation and has an IR spectrum with a strong broad -OH peak in the 3500–3000 cm^−1^ range that will cover up a lot of other substances peaks, such as amides, lipids, proteins, and alcohols, and makes it difficult to identify and interpret spectra. Moreover, water is not a very good solvent for IR samples, or glass (in the form of cuvettes or optics), which significantly increases the attractiveness of using near-IR spectroscopy in biology [4].

In addition, cheaper sensors and the development of a mathematical apparatus for analyzing and interpreting spectra have also contributed to the increased popularity of near-IR spectroscopy, though mid-IR spectroscopy, due to its higher sensitivity and simpler processing and interpretation of spectra, still remains more popular.

#### 3.3.2. IR Radiation Sources

The sources of continuous IR radiation usually represent so-called “blackbody” sources because the temperature dependence on the wavelength in such a source has the form of the radiation curve of the absolute black body [32]. Such blackbody sources include, for example, a silicon carbide rod called the Globar source (1:1 Si and C) or the Nernst rod made from the oxides of rare earth elements, Y_2_O_3_ and Er_2_O_3_. The maximum intensity of such radiation sources lies in the region of ~5000 cm^−1^ (~2 μm), and they have a rather wide wave range. In addition, the use of semiconductor lasers (or laser complexes) as IR radiation sources has been developed in recent years, though currently they are rather rarely used.

#### 3.3.3. Design of IR Spectrometers and Leading Varieties of IR Spectroscopy

One of the main stages in the development of IR spectroscopy was the appearance of IR spectrometers based on the Fourier transform, which made it possible to simultaneously evaluate the effect of a radiation on the sample within the whole studied range (before this moment, scientists used devices equipped with a broadband light source and a monochromator separating certain wavelengths to assess their effects) [5]. Usually, the Fourier Transform Infrared (FTIR) spectrometers are based on the Michelson interferometer, in which the signal received from a sample and the control signal are compared between themselves and all the frequencies of the transmitted light are modulated on the control beam using a movable mirror. Then the resulting signal difference is subjected to a Fourier transformation, resulting in a dependence of the transmission intensity on the wavelength (wavenumber). This approach significantly reduces the requirements for the detector and the cost of the device while maintaining a high quality of spectra. Currently, the vast majority of IR spectrometers used in the near- and mid-IR range represent Fourier IR spectrometers.

According to the energy conservation law, the energy of light (I_0_) influencing a sample can be written in the following form [31]:I_0_ = I_A_ + I_T_ + I_R_ + I_S_,(6)
where I_R_ is the light reflected from a sample; I_A_ is the absorbed light; I_T_ is the transmitted light; and I_S_ is the light scattered on a sample. In IR spectroscopy, information about the vibrations of a molecule can be obtained by determining I_A_ at a certain frequency.

There are several modes of the I_A_ measurement based on the registration of the I_0_ value and one of the remaining indices (I_T_, I_R_, or I_S_); note that the remaining part of the radiation should tend to zero, which represents the main goal of a sample preparation.

The most common method of obtaining information about molecular vibrations is a transmission mode. This technique is used for the study of liquid or, rather, thin solid samples. Sometimes powdered samples are pressed into tablets containing materials that are optically transparent in the IR region (KBr, NaCl, LiF, CsI, etc.) to reduce light scattering and increase the transparency of samples in the IR range. Transmission spectroscopy is very often used to perform measurements in thin films or biological objects (tissues, culture samples, etc.) [33].

However, in recent times, the most popular measurement techniques have been based on the registration of the reflection using either external (mirror) reflection, when light spreads in a medium with a lower optical density, such as the air, or internal reflection, when light spreads in an optically dense medium [29].

The most common modification of IR spectroscopy is based on the registration of reflection on the surface of a prism made of materials with a high refractive index (Figure 3a). This modification is called Fourier transform infrared spectroscopy with attenuated total reflection (ATR-FTIRS) [31]. In this case, a tested sample is tightly pressed to the surface of a prism made of a material with a high refractive index. IR radiation is completely reflected from the interface between the prism and the sample, but it induces a local radiation (evanescent waves) that penetrates into the sample to a small depth and is absorbed by sample molecules. The IR beam is multiplicatively reflected from the surface of the prism-sample interface, while the absorption signal is summed up, which makes it possible to obtain a quite intense signal. This measurement method is very sensitive to the sample uniformity and texture (presence of a polished reflective surface), although in some cases measurements are also possible on a very thin layer of a highly dispersed homogeneous powder or liquid.

Sometimes it is impossible to obtain a homogeneous, smooth sample with a reflective surface or a sample compressed into a dense tablet. An example of such a sample is powder. In this case, in addition to reflection, a diffuse scattering and/or absorption of radiation by a sample is observed, which requires the use of special techniques. To work with such substances, one can use diffuse reflection techniques, such as Diffuse Reflection Infrared Fourier Transform Spectroscopy (DRIFTS) [31]. This method is based on the registration of radiation scattered by a thick layer of a powdery sample (Figure 3b). This method is very demanding in relation to the sample preparation. To obtain acceptable light scattering values, it is necessary to have an amorphous powdery preparation as well as to dilute the sample with a substance characterized by optical transparency in the IR region.

#### 3.3.4. IR Microspectroscopy

To date, various constructions of IR microspectrometers (IR microscopy) have been designed, which provide the obtaining of IR images of samples, local measurement of IR spectra under various modes, and evaluation of local changes occurring in samples [34]. The result of the microspectrometer’s use is a bitmap image (a hypercube), in which each pixel is associated with the spectrum registered in this region. These images are obtained either by measuring the whole spectrum at each point of the sample (dispersion systems), or by simultaneous registration of all local spectra of a sample (imaging systems) [31,34]. The application of such a technique requires the use of special lenses. In general, as in the case of other optical microscopes, its lateral resolution is determined by a diffraction limit (0.61λ/NA ratio, where NA is the aperture of the used lens and λ is the wavelength of the light source). Since the wavelength in the IR region exceeds the wavelength of the visible range, the lateral resolution of IR microscopes is lower than that of “traditional” ones, which use sources of visible light, and is determined by the wavelength of the light used (>6 μm [3,34]); at the same time, IR microscopes are capable of estimating samples with a higher thickness (~5–50 μm in the transmission mode). In the case of working with biological samples, they have the same advantages and disadvantages as conventional IR spectrometers, including restrictions for working with aqueous solutions. In recent times, various types of IR microspectrometers have been quite actively used for tumor diagnosis, including in vivo conditions [35]. However, this type of IR spectroscopy is rarely used for solving environmental tasks.

#### 3.3.5. The Effect of Water and Atmospheric Carbon Dioxide on IR Spectra

IR spectra (especially mid-IR spectra) are largely absorbed and influenced by the presence of water and CO_2_ (in the case of air measurements). Water, as a dipole molecule, actively absorbs IR radiation and has an IR spectrum with a strong broad -OH peak in the 3500–3000 cm^−1^ range and a less wide -OH peak in the region ~1650 cm^−1^ (in addition, there are several less intense peaks in the spectrum of water, e.g., 2100, 710–645 cm^−1^, as well as 3250 cm^−1^ overtones) that will cover up a lot of other substances peaks, such as amides, lipids, proteins, and alcohols. This makes it difficult to identify and interpret spectra; (furthermore, water is not a very good solvent for IR samples). This does not really matter in the case of NIR spectra, where the overtones of –OH groups do not make such a significant contribution as with mid-IR spectroscopy. In the case of mid-IR spectroscopy, special sample preparation is used to minimize the effect of the water signal; in addition, using infrared spectroscopy with attenuated total reflection allows recording a signal in a thin layer of a substance, which also reduces the absorption of water. As a rule, when registering a signal in aqueous solutions, the spectrum of water is subtracted from the spectrum of the sample.

In the case of biological experiments, the influence of CO_2_ molecules, unlike water, is not as significant due to its low concentration in the air. In addition, the most intense band of the IR spectrum of CO_2_ (~2300–2400 cm^−1^) does not overlap with the bands of the spectra of most biological molecules.

#### 3.3.6. Using IR Spectroscopy for the Study of Biological Objects

To date, different types of IR spectroscopy have become powerful and universal tools for the study of biological objects due to their non-invasiveness, possibility to assess molecular groups and molecules, relatively high speed of measurements, applicability to a large range of samples, and provision of both qualitative and quantitative information. Currently, a database of spectra has been developed, which makes it possible to immediately identify a large number of functional groups and compounds and provides a large amount of experience in the processing and analysis of the measured spectra for various applied fields. The main disadvantage of the IR spectroscopy methods when working with biological objects is the high absorption of IR radiation by water molecules. This significantly reduces the possibility of using IR spectroscopy methods for the work with aqueous solutions, so IR-transparent materials are used to register IR transmission; often, samples are prepared by tableting with potassium bromide, which practically does not absorb in the IR range [4].

### 3.4. Raman Spectroscopy

Unlike IR spectroscopy, which represents a kind of light absorption, Raman spectroscopy is based on light scattering. However, unlike Rayleigh scattering, light scattering in this case is inelastic (light and matter can exchange their energy, which leads to a change in the scattered light frequency) [29].

#### 3.4.1. Stokes and Anti-Stokes Scattering

Scattering can be considered a very fast process of photon absorption and emission. Absorbing a light photon, a molecule turns into an unstable excited electronic state (virtual level), from which it emits a photon (Figure 2c) [29]. In the case of Rayleigh scattering, a molecule absorbs a photon at the zero vibrational level and returns to the same level after radiation (no energy change occurs). In the case of Stokes scattering, a molecule absorbs a photon at the zero vibrational level and then emits only a part of the energy of the absorbed photon, thus transiting to the first vibrational level with a simultaneous decrease in the scattered light frequency (wavenumber) (Figure 2c). In the case of anti-Stokes scattering, a molecule absorbs a photon being at the excited (non-zero) vibrational level, and after radiation it loses a part of its energy and transits to the zero vibrational level with the simultaneous increase in the scattered light frequency (wave number). The population of higher vibrational levels is significantly lower than that of zero level, so the intensity of anti-Stokes lines in the Raman spectrum is significantly lower compared to that of Stokes lines. Usually, the more intense Stokes part of the spectrum is considered the Raman spectrum.

#### 3.4.2. Raman Shift

The results of measurements performed by Raman spectroscopy are usually presented as the dependence of the scattered light intensity on the frequency shift, also called the Raman shift (Δ$\overset{-}{\nu }$) expressed in cm^−1^. The frequency shift is a difference between the Rayleigh scattering frequency (the frequency of the radiation source) and the frequency of the corresponding Stokes line. In terms of wavelengths, the frequency shift will look like this:(7)$$\overset{-}{\nu }=1/\lambda_{R}-\lambda_{S}$$
where λ_R_—is the wavelength of a radiation source λ_S_—is the wavelength of the Stokes line. The data presentation using such a scale is more convenient since it is directly related to energy (see Formula (2)); in addition, it allows one to compare the Raman spectra obtained on the equipment with different radiation sources without shifting the obtained results.

#### 3.4.3. Photoluminescence

The intensity of the Raman spectrum lines is extremely low; it is significantly (by several orders of magnitude) less than the intensity of lines from both Rayleigh scattering and light absorption processes [30]. When a molecule absorbs a photon and transits to the excited electron state (here we do not mean IR adsorption), it radiates a photon (usually characterized by a lesser frequency) and transits to the fundamental state. This process is called photoluminescence (Figure 2b) [29]; depending on the configuration of the excited electronic state, it is divided into fluorescence and phosphorescence. The photoluminescence spectrum may overlap with the Raman spectrum, which is especially relevant when working with pigment-containing cells and samples since it significantly complicates the recognition of lines and the interpretation of spectra. One of the most common options for providing reduction of photoluminescence is the use of a radiation source working in the near-IR range, which does not cause a significant photoluminescence of biological samples (at the same time, this can also result in a decreased intensity of the Raman spectrum bands) [29].

#### 3.4.4. Main Components of Raman Spectrometers

Raman spectrometers should have a good light source. Usually, they use continuous lasers characterized by high stability and a narrow bandwidth. Another important component of the Raman spectrometer is a filter that separates a wanted signal containing the Raman spectrum lines from the Rayleigh signal of the radiation source. Since the intensity of the Raman spectrum bands is very low, it is very important to have the possibility of a clear separation of the unbiased light from the wanted signal with as low losses as possible. Other components of spectrometers usually do not fundamentally differ from spectrometers working in the visible region; both dispersion and Fourier spectrometers can be used [36]. In recent times, portable Raman spectrometers have been actively used, though they do not possess such good characteristics as stationary spectrometers but are very mobile [37].

#### 3.4.5. The Spectral Range

The spectral range of Raman instruments is mainly determined by the light source used; depending on the wavelength, the studied range can vary from 250 to 4000 cm^−1^ (UV, visible, near-IR, and mid-IR ranges).

#### 3.4.6. Resonance Raman Spectroscopy

If the energy (wavelength) of the incident light is close to the energy of the electronic transition of the studied bond (molecule), i.e., the resonance occurs, then a sharp increase in the intensity of the corresponding spectral lines by several (up to six) orders of magnitude is observed [38]. In the case of large molecules having several functional groups, only those group frequencies that are associated with an electronic transition (absorption) in the wavelength region of exciting light are amplified.

In practice, there is a great chance of achieving the resonant Raman spectrum of a compound (if it theoretically has a Raman spectrum) if a wavelength close to the maximum of the absorption spectrum of this compound is chosen. Among the frequently occurring biological compounds providing such a chance, one can mention various carotenoids or polyenes, chromophores, hemes, chlorophylls, etc.

#### 3.4.7. Surface-Enhanced Raman Spectroscopy (SERS)

This approach is used to amplify the Raman spectroscopy signal. This goal is usually achieved by the addition of colloids of precious metals (gold or silver) to the sample or the use of nanostructured substrates made of precious metals, on which the sample is applied [39]. The light influence on the surface of metal particles results in an excitation of plasmons on the metal surface that, in turn, leads to an increase in the surface electric fields and a significant (up to 10^11^) increase in the intensity of Raman spectrum bands near the metal surface.

There are many variants of this method, from the simplest ones (a sample is added to a test tube containing colloids, and then the measurement is performed) up to serious modifications of the equipment and measuring techniques (reviewed in [40]). This method is not universal and requires selecting conditions for each measured object. It is also very sensitive to the sample preparation and measurement conditions, which significantly reduces its value as a tool for routine environmental measurements. Nevertheless, such an approach makes it possible to perform measurements using standard Raman spectrometers even if the amount of the studied material is rather insignificant, which is a significant advantage of this technique.

#### 3.4.8. Raman Microspectroscopy (Raman Microscopy)

The basic principles of IR and Raman microspectrometry mainly coincide. In the case of Raman miscospectrometry, there are also two approaches for obtaining images; however, the work with biological samples, especially under in vitro or in vivo conditions, is arranged using dispersion microscopes since they provide a local and more gentle effect on a sample that is much more important for the Raman microscopy than for the IR microscopy due to the use of light sources with shorter wavelengths. Like in the case of other optical microscopes, the lateral resolution of Raman microscopes is determined by the diffraction limit and is 0.3–0.5 microns [3,34]. The axial resolution also depends on the wavelength and aperture (2nλ/(NA)^2^, where n is the refractive index of the medium [34]) and does not exceed 10 microns. In the case of work with biological samples, Raman microscopes are generally characterized by the same advantages and disadvantages as conventional Raman spectrometers. It is possible that the spectra measured using this equipment are less affected by unwanted noise due to the use of an objective; for the same reason, the microscope has a more pronounced photodestructive effect.

Currently, Raman microscopes represent quite complex, expensive, and poorly transportable devices (of course, with some exceptions) that significantly limit the scope of their application.

#### 3.4.9. Other Types of Raman Spectroscopy

Today there is a large number of different Raman spectroscopy types [41] as well as approaches combining the Raman spectroscopy with some variants of the probe microscopy (such as atomic force or tunneling microscopy) or the scanning electron microscopy. There is also a combination of Raman microscopy with optical tweezers [2].

Sometimes the TERS (Raman Spectroscopy with tip amplification) method based on the use of a probe microscope and a special gold or silver tip is used; the use of such tips enhances the Raman signal (the lateral resolution corresponds to the tip size equal to ~30 nm) while simultaneously obtaining topographic data by a probe microscope [41].

The use of a pulsed laser (or several pulsed lasers) also makes it possible to expand the capabilities of traditional Raman spectroscopy [41]. For instance, stimulated Raman scattering (SRS) is utilized to enhance the intensity of the spectrum (or individual frequencies of the spectrum). This approach is based on the transition of a molecule to an excited state and the use of two lasers, the frequencies of which are selected in such a way that their difference would correspond to the Stokes frequency. Hyper Raman spectroscopy (HRS) is used to reveal frequencies that are typically not visible in the Raman spectrum, while Coherent anti-Stokes Raman spectroscopy (CARS) uses two pulse lasers to generate the anti-Stokes part of the Raman spectrum, allowing for information to be obtained when working in a fluorescent medium [41].

In addition to the above-described approaches, there are many other types of Raman spectroscopy, but all of them are quite specific, require significant modifications to standard equipment, and are generally unsuitable for in vitro, in vivo, or even in situ studies [41].

Thus, vibrational spectroscopy, which encompasses near-, mid-IR, and Raman spectroscopy, allows for the evaluation of the presence, conformation, and concentration of various substances in biological objects across a broad range using vibrational spectra. The peculiar properties of near-, mid-IR and Raman spectroscopy are summarized in Table 2.

## 4. Practical Application of Various Types of IR and Raman Spectroscopy in Biological and Environmental Studies

### 4.1. Application of Raman Spectroscopy

Since the second half of the 20th century laser-excited Raman spectroscopy as fast non-invasive and slightly damaging method has been widely used in investigation of different biological objects. During this time, in general, two main problems can be formulated for the use of Raman spectroscopy to study biological samples: (*1*) very low signal intensity of the Raman spectrum bands that requires enhanced laser power and a well-developed mathematical apparatus for treatment of spectra; and (*2*) rapid damage of radiated samples that requires a very careful selection of the radiation power and duration for each biological object. Unfortunately, due to these peculiarities, not all types of Raman spectroscopy are effective when working with biological objects. The solution of these problems is critical for the successful use of Raman spectroscopy. However, the aforementioned problems do not prevent the use of different methods of Raman spectroscopy for various biological and environmental studies.

Table 3 shows some examples of substances and objects that were detected using different methods of Raman spectroscopy.

#### 4.1.1. Evaluation of Carotenoids

One of the most promising ways to use Raman spectroscopy for the study of biological objects is a quantitative and qualitative evaluation of carotenoids.

Carotenoids play a very important role in the majority of procaryotic and eucaryotic microorganisms. Being antioxidants, they protect organisms from the damage caused by UV and other types of radiation and facilitate the reparation of damaged cells; in the case of plants, carotenoids are also components of the photosystem [77,78]. Carotenoids are common natural pigments that usually have a reddish-yellow color and are unsoluble in water. Due to their antioxidant activity, they are widely used in medicine and the food industry; some of them are vitamin A precursors [78]. As a rule, carotenoids belong to polyenes with conjugated double bonds; the vast majority of them are tetraterpenoids consisting of eight isoprene units (the structure of a single isoprene unit is shown in Figure 4a). These compounds may contain acetylene and diene bonds, as well as hydroxyl, cyclic, glycoside, sulfo-, keto-, and other groups [78].

Starting from the early appearance of Raman spectroscopy, carotenoid molecules became very popular objects of study, which is explained by the fact that, in the case of a light source with the emission range of 460–540 nm, they show a very intense resonance signal with the maximum near 470 nm (or 515 nm for lycopene) [79]. The Raman spectra of carotenoids are usually very similar to each other (Figure 4) and are characterized by three clearly distinguishing lines: ν_1_ (~1530 cm^−1^), whose intensity and position are determined by the stretching modes of conjugated C=C bonds; ν_2_ (~1120–1200 cm^−1^), whose intensity and position are determined by the mixture of C=C and C–C bond stretching modes with C-H bending modes; and ν_3_ (~1000 cm^−1^), whose intensity and position are determined by the stretching modes of bonds between the main chain and the side methyl carbons [77]. In some cases, the ν_4_ line (~950 cm^−1^) associated with the out-of-plane C-H modes is also taken into consideration. Based on these three basic bands and evaluating the positions (distances between the peaks) and intensity (area) of their peaks, one can assess a number of parameters [43].

The presence of microorganisms in the studied sample can be detected by the presence of carotenoids in the Raman spectrum of the sample; for example, the presence of carotenoids can be used to detect extremophiles [42]. Not all extremophiles contain carotenoids, but if their colonies include carotenoid-containing cells (like some carotene-containing procaryotes or eucaryotes), the detection of such cells becomes much easier.

Carotenoids in microalgae are divided into primary and secondary ones [42]. Primary carotenoids (such as β-carotene, lutein, and lycopene) are involved in the photosynthetic processes, while secondary ones (for example, astaxanthin) are synthesized under various stress conditions and usually stored in lipid droplets that make it possible to monitor the stress development [44]. Thus, even the qualitative registration of an increase in the content of carotenoids and/or their redistribution in a single cell may indicate stress development.

Different carotenoids are characterized by slightly differing maximums of lines and ratios of their amplitudes. For example, the maximum of the ν_1_ band for astaxanthin and neoxanthin is slightly shifted to the long-wave range, while the corresponding maximum of lycopene and bacterioruberin is shifted to the short-wave range [49]. The original method of lycopene assessment in the carotenoid mix was proposed in [45]. The absorption spectrum of lycopene is characterized by an additional maximum at 514.5 nm that makes it possible to evaluate the presence and content of this compound using a laser emitting at the corresponding wavelength. Based on the shift of the ν_1_ band registered by Raman spectroscopy, a method for measuring hydration level of Arctic lichens was proposed [46,47]. This parameter is important since hydration activates photosynthetic processes in lichens.

Based on the ν_1_/ν_2_, ν_1_/ν_3_, and ν_2_/ν_3_ peak ratios as well as on other methods, it was shown that pH changes in algal cells did not cause changes in the content of chlorophyll *a* and *b* [48], whereas the content and conformation of carotenoids seemed to be changed. Studies intended to evaluate the effect of a thiamethoxam insecticide on maize leaves showed this compound caused changes in the photosynthetic activity of plants [50,51]. Based on the analysis of ν_1_/ν_2_, ν_1_/ν_3_, and ν_4_/ν_3_ ratios, authors concluded this effect is accompanied by changes in the conformation of carotenoids but not in their content.

Using semi-quantitative peak intensity evaluation for the Raman spectra of carotenoids, authors of other studies determined the maturity degree of watermelons [54] and the strength (maturity) of hot peppers [55].

Raman spectroscopy can be used for a non-invasive assessment of the total content of carotenoids in a sample (cells). For this purpose, the area (intensity) of the ν_1_ band [80] (sometimes normalized to the intensity of other bands [52]) is used.

Thus, the use of Raman spectroscopy provides a rapid, non-invasive (in vivo) detection of carotenoids as well as an approximate estimation of their composition and content in single cells and cell cultures. However, such a purpose requires a certain approach and set of requirements for the equipment and measurement technique. According to almost all authors, signals obtained from living cultures have a significant noise component provided mainly by fluorescence. Noise reduction sometimes becomes a non-trivial problem requiring a special approach to both equipment and methods [81]. However, it is strongly required to provide a correct estimation of the peak intensity, position, and even maxima. In addition, unlike HPLC, the difference between the peaks of different carotenoids is usually not too large to significantly complicate the accurate identification of carotenoid types in a sample [42]. On the other hand, the use of Raman microscopy makes it possible to perform measurements on single cells, which can be useful in the case of heterogeneous cultures. In general, such a conclusion is rather universal and can be applied not only for carotenoid studies but also for the studies of other pigments and compounds by Raman spectroscopy.

#### 4.1.2. Assessment of Non-Carotenoid Pigments and Photosynthesis

Biological pigments represent a large group of naturally pigmented compounds produced by living organisms. Along with carotenoids, this class of compounds includes such groups as quinones, flavonoids (polyphenols), chlorophyll, pheophytin, cytochromes, hemoglobin, etc. Raman spectroscopy has been quite actively used for a long time to assess the presence and content of such pigments in various microorganisms (algae, bacteria, lichens, and endoliths, including those living in halites; reviewed in [43]).

The Raman spectroscopy was also used to study microbial colonies for the presence of such pigments as flexirubin, which has a carotenoid-like structure, scytonemin (from cyanobacteria), chlorophylls, etc. [43,57,58].

#### 4.1.3. Use of Raman Spectroscopy in Biofuel Production

One of the most important directions for the development of current biotechnology is biofuel production from raw materials of animal or plant origin, microbial products, or organic wastes. Biofuel can be divided into three categories: (*a*) gas fuel (biogas and hydrogen); (*b*) solid biofuel (fuel briquettes produced from wood processing waste, firewood, etc.); and (*c*) liquid (transport) biofuel, such as bioethanol and other low-molecular alcohols, biodiesel, etc. In this case, biodiesel represents a liquid fuel consisting of a mix of monoalkylated esters of fatty acids and produced from triglycerides (oil) and fatty acids by the transesterification reaction [61]. The sources of liquid biofuel include agricultural crops characterized by a high content of lipids (for biodiesel production), starch, and sugars (for bioethanol production) [64]; wood processing waste, grass, and straw containing a large amount of cellulose and lignin (for bioethanol production) [59]; and algae and microalgae, which are considered now as the most promising sources [60]. As a rule, the oils and fatty acids contained in algae are used for biodiesel production, though they can also be used for bioethanol and biomethane production.

Raman and infrared spectroscopy are used not only for the evaluation of a raw material (usually algae) for its readiness for biofuel production but also during different production stages. Based on Raman microscopy, equipment capable of contactless monitoring of a biofuel production process has been developed [59,60,64]. For example, the study [64] describes the Raman-based evaluation of the process of esterification of the rapeseed oil into a biodiesel (Figure 5). The basic peak, whose intensity was used as a marker of the rape oil conversion into the ester, was the 1655 cm^−1^ peak corresponding to the valency oscillations of the C=C stretching mode and typical for unsaturated fatty acids.

The equipment capable of measuring and controlling the lignin and cellulose conversion to bioethanol is described in [59]. The main component of this equipment was a Raman spectrometer. The assessment of the components inside a reactor was based on the evaluation of peaks typical for the Raman spectra of ethanol (a typical and very strong peak at 883 cm^−1^, whose intensity is associated with C–C stretching vibrations), glucose (an intensive peak at 1120 cm^−1^), glucose oligomers and polysaccharides (a typical glycosidic bond peak at 920–960 cm^−1^), and some other compounds (fructose, glycerin, cellulose, etc.); additionally, a specially developed mathematical simulator was used.

Today, the Raman spectroscopy-based evaluation of the content of lipids (unsaturated fatty acids) in algae and microalgae has become very popular, which is explained by the most efficient use of these living organisms in biodiesel production (the lipid content in algal cells may reach 71%) [83]. For this purpose, scientists not only evaluate samples containing a large number of cells but also apply Raman microscopy to evaluate single cells. Using Raman microscopy, one can non-invasively and rapidly identify algae by the presence of Raman peaks typical for oils and fatty acids (these peaks include mainly 1650 and 1440 cm^−1^ indicating unsaturated and saturated bonds, respectively), determine the presence and content of lipids in algae, and identify the unsaturation level of fatty acids composing lipid droplets in algae [60]. For example, the study [62] demonstrated a possibility to evaluate the ratio of unsaturated and saturated lipids in algal cultures by the ratio of intensities of the 1650 cm^−1^ and 1440 cm^−1^ peaks. The first peak is associated with the valency oscillations of C=C bonds, whereas the second one is associated with the deformation (bending) oscillations of –CH_2_ bonds. As the intensity of the first peak increases, the intensity of the second peak decreases.

#### 4.1.4. Evaluation of the (Micro)algae Composition

One of the most relevant fields of application for Raman spectroscopy (sometimes combined with other techniques and approaches) includes identification of compounds and evaluation of their intracellular distribution (mapping), content, and conformation in algae and microalgae. This allows a researcher to use algae in environmental studies; as a rule, a small number of algal cells is sampled from a bioreactor or their habitat and then placed under a microscope. The work with single cells requires a rather large number of measurements to obtain reliable data but, at the same time, provides some additional possibilities (e.g., the above-mentioned intracellular mapping of compounds); most importantly, it makes it possible to work with mixed cultures but evaluate only the selected cells.

The most common variants of this type of analysis include the localization of various carotenoids and other pigments as well as the evaluation of the content and localization of compounds contained in lipid bodies (see, for example, [42,49,52,56,60,62,84,85]). Some of these studies are already mentioned in other sections of this review.

Another important task that can be realized by Raman microscopy is the control of inorganic polyphosphate inclusions in cells. Phosphorus is one of the important elements required for the cell’s functioning. Its availability is limited for the majority of living things on Earth; at the same time, the reserves of this compound can be periodically replenished, which determines the relevance of its intracellular storage (usually in the form of inorganic polyphosphates). The use of Raman microscopy makes it possible to estimate the number and location of inorganic polyphosphate inclusions (1160 cm^–1^ peak) [65,66].

Nitrogen is also important for the cell’s functioning. Crystalline guanine (C_5_H_5_N_5_O) inclusions are known to be one of the important forms of nitrogen storage in phytoplankton. The use of Raman microscopy makes it possible to estimate the content and intracellular localization of guanine [67].

To date, the method of a simultaneous evaluation of several compounds in algal cells by Raman microscopy has also been developed. The following spectral ranges are used for such evaluation: 457–507 cm^−1^ for starch (with the maximum near 479 cm^−1^), 2836–2869 cm^−1^ for lipids (maximum at 2854 cm^−1^), 1143–1190 cm^−1^ for polyphosphates (maximum at 1159 cm^−1^), 613–684 cm^−1^ for guanine (maximum at 651 cm^−1^), and the 2795–3060 cm^−1^ region for hydrocarbon compounds characterized by the presence of covalent C–H bonds (maximum at 2935 cm^−1^) [68]. In this case, the reduction of autofluorescence, which is generated by the photosynthetic apparatus of algae and whose value can exceed the registered signal, is especially important. Moreover, under certain conditions, the Raman signal of some compounds (e.g., carotenoids) is able to overlap the signals of other substances. All these problems require some special procedures (photobleaching under special conditions), allowing the registration of the spectra of such substances as starch, lipids, inorganic polyphosphates, etc. [84].

#### 4.1.5. Assessment of Shells Using Raman Spectroscopy

One of the interesting applications of Raman spectroscopy in the field of biomonitoring is the assessment of the size and mineralization level of shells (exoskeletons). The composition of shells and exoskeletons of marine animals is closely related to the ocean acidification problem, i.e., the pH decrease resulting from the adsorption of atmospheric carbon dioxide by the ocean. Such acidification results in the demineralization of marine animals and the failure of their ability to form shells and exoskeletons consisting of various crystalline forms of calcium carbonate. This, in turn, influences food chains based on such animals and may cause serious environmental problems. The consequences of ocean acidification primarily concern marine species, whose shells are formed of calcium carbonate [86].

The shell composition can be assessed by Raman spectroscopy. As a rule, shells contain polyenes (carotenoids or similar pigments), various forms of calcium carbonate (amorphous and crystallic, such as calcite and aragonite) [69,70,71], and magnesite [72]. Aragonite is a part of the nacre layer, which represents the internal shell layer, so the appearance of the aragonite spectrum during in vivo analysis of a shell indicates shell demineralization resulting from ocean acidification [70]. Note that the Raman spectra of aragonite and calcite are quite similar to each other, though they can be distinguished during the analysis. The calcite spectrum is characterized by the peaks at 156, 282, and 1087 cm^−1^, while the aragonite spectrum includes peaks at –155, 206, and 1086 cm^−1^ [73]; in the case of magnesite, there is a band at 1089 cm^−1^ [72]. The peak at 1085–1087 cm^−1^ is rather large, and it is difficult to separate it from the peaks typical for aragonite and calcite; due to this fact, it is considered the manifestation of the presence of different carbonate forms [71].

To date, the equipment has been developed, which provides a possibility to evaluate both in vivo and ex vivo calcification levels and the mode of biomineralized structure of various organisms even under deep-water conditions [72,74,87]. Thus, Raman spectroscopy provides a non-invasive in vivo and ex vivo evaluation of the biomineralization of shells and exoskeletons of marine animals.

#### 4.1.6. Use of Surface Enhanced Raman Spectroscopy (SERS)

The advantages of this method include the possibility to work with much lower concentrations than Raman microscopy and Raman or resonance Raman spectroscopy, while the drawbacks include the necessity to use special nanoparticles or nanostructured substrates usually made of precious metals and the special methods of their application. In spite of the great potential of SERS and its use in other fields of science (reviewed in [1,2,88,89]), this approach (or approaches with the potential for SERS use) is not commonly used in environmental studies. Nevertheless, such studies are carried out, and their number is quite large (reviewed in [40]).

One of such studies demonstrated the possibility of using SERS combined with further mathematical analysis to evaluate the level of banana infection with *Fusarium oxysporum* f. sp. *cubense*, a fungus causing Fusarium wilt of banana [90].

Another study [53] included the use of SERS for evaluation of the content of a fucoxanthin carotenoid in some algal cultures by the peak intensity at 1520 cm^−1^; the sensitivity of the method was ~1 × 10^−6^ M. Compared to the “traditional” variants of Raman spectroscopy, the use of SERS provides at least a two-order extension of the range for such evaluation.

Authors of another study [63] analyzed the possibility of using SERS to evaluate the presence of fatty acids and their saturation/unsaturation level as well as the presence and intracellular localization of other compounds (chlorophyll, proteins, carbohydrates, and carotenoids) under conditions of anoxia and also under different pHs. The basic peaks chosen for evaluation and mapping of fatty acids were 1621 cm^−1^ (which corresponds to the cis C=C bonds and serves as the marker of saturated bonds) and 1337 cm^−1^ (which corresponds to the CH_2_ scissoring and represents the marker of unsaturated bonds). The study was carried out using the Scenedesmus quadricauda CASA CC202 culture.

Also, SERS is used for ultrasensitive bacteria (pathogen) detection, e.g., *E.coli* or *S. typhimurium* [75,76]. This is a rather complex technological problem requiring the use of special particles containing biorecognition molecules (e.g., antibodies and aptamers) [91] that specifically bind to cells. As a result, it becomes possible to detect these pathogens rapidly and with high sensitivity.

### 4.2. Application of Different Types of IR Spectroscopy

As we have already mentioned, the advantages of IR spectroscopy include noninvasiveness, non-damaging or slightly damaging effects (even compared to the Raman spectroscopy), relatively rapid obtaining of results, and (unlike the Raman spectroscopy) the possibility to evaluate polar molecules (alcohols, phenolic compounds, etc.) [4]. As a result, different types of IR spectroscopy are quite actively used for the monitoring of environmental pollution as well as for solving different biomedical and other tasks, such as tumor diagnostics (reviewed in [4,5,92,93]). Moreover, IR spectroscopy is a cheap method compared to its analogues [5]. A study of biological samples usually requires the use of near-infrared (1000–2500 nm, 10,000–4000 cm^−1^) and mid-infrared (2500–25,000 nm, 4000–400 cm^−1^) spectroscopy [4]. Unfortunately, the low sensitivity of near-infrared (NIR) spectroscopy and the limitations of mid-IR spectroscopy in the case of dealing with aqueous solutions, as well as the lower resolution of the method compared to Raman spectroscopy [3], result in significant problems with microscopic studies [4] that limit the potential of the method in relation to living objects and environmental studies.

The result of measuring biological samples is an IR spectrum, which, like in the case of Raman spectroscopy, represents a superposition of numerous overlapping peaks, whose presence and intensity depend on the chemical bonds in the studied compounds and their conformation. Based on the presence of typical chemical groups, it is possible to determine the presence of proteins, amino acids, fatty acids, polysaccharides, phenolic compounds, etc. However, such a large number of peaks significantly complicates their interpretation and requires the use of special mathematical apparatus, which (in contrast to Raman spectroscopy) has already been developed. The obtained spectral data are treated using various multiparametric analysis methods, such as principal component analysis (PCA), soft independent modeling of class analogy (SIMCA), hierarchical clustering analysis (HCA), canonical variation analysis (CVA), factor discriminant analysis (FDA), partial least squares discriminant analysis (PLSDA), and various regression models, including multiple linear regression (MLR), principal component regression (PCR), partial least squares regression (PLSR), etc. [33]. When applying these methods, there is no need to use the values of individual peaks; in these cases, whole (or selected sections of) IR spectra can be used without isolating individual peaks.

Table 4 shows some examples of using different methods of IR spectroscopy for environmental studies.

All the aforesaid advantages of IR (especially NIR) spectroscopy, as well as the availability of relatively inexpensive and cheap equipment, make it possible to perform environmental studies using IR spectroscopy (reviewed in [6]). Since IR and especially NIR spectroscopy provide the possibility of working with aqueous solutions, this method allows for the in vivo qualitative and quantitative assessment of the composition of animals and plants and the evaluation of changes occurring in them. The objects of IR spectroscopy may include single-celled fungi, lichens, microalgae, plant and animal tissues (membranes, bones, plant organs), multicellular plants and animals, and products of their vital activities (consumed food, excretions, feces, etc.) [6].

#### 4.2.1. Evaluation of Various Active Compounds in Plants

NIR and IR spectroscopy is often used to evaluate the antioxidant potential of extracts or raw materials from various plants (see [94,95]). Such an effect can be obtained by evaluating the presence of compounds with antioxidant activity (for example, various phenolic compounds) based on their characteristic peaks. In addition, the use of various computer models along with these typical peaks makes it possible to identify these substances and evaluate them not only qualitatively but also quantitatively.

For example, IR spectroscopy allowed researchers to register changes in the composition and content of flavonoids, organic acids, polysaccharides, essential oils, and other compounds during the harvesting of Japanese honeysuckle [96].

Along with phenolic compounds, IR spectroscopy provides a possibility to evaluate such substances as triterpenoids [97], various polysaccharides, etc. [98]. NIR and IR spectroscopy are also applied for the study of polysaccharides, organic acids, and carotenoids in various fruits [99,100].

The use of IR and Raman spectroscopy for plant cuticle measurement is reviewed in [101]. The plant cuticle represents a membrane on the surface of terrestrial plants that is impregnated with various substances (cutin, cutane, waxes, polysaccharides, and phenolic compounds) and whose main function is to prevent significant water losses and water intake from the environment. Vibrational spectroscopy allows evaluating the above-mentioned compounds as well as the state and thickness of a cuticle, its changes during the life of plants (including fossil plants), and the interaction of the mentioned compounds with exogenous molecules.

Due to its noninvasiveness, the possibility to work with various substances, and a significant penetration depth exceeding that for IR and Raman spectroscopy, NIR spectroscopy is often used to control different characteristics of wood [104] and evaluate the chemical structure and physical properties of samples as well as changes in their composition and degradation level (including in vivo changes). For example, IR spectroscopy is used to evaluate the content of lignine and phenolic compounds in the wood [105,106]. Today, NIR spectroscopy is also used for the evaluation of a sample’s humidity [107]. Using IR spectroscopy and the corresponding mathematical simulators, it is possible to evaluate the wood density based on the ratio between cellulose, lignine, and other wood components [108]. Note that wood characteristics can be evaluated using not only IR spectra and/or spectrum-based simulators but also sample images obtained by IR spectroscopy [104].

#### 4.2.2. Biodiesel Quality Control

The NIR and mid-IR spectroscopy are used to control the quality of biodiesel at all stages of its production and application (choice of the raw materials and reduction of their cost, trans-esterification reaction, and determination of the properties, quality, and contaminants of biodiesel) [61]. For example, the presence of a large amount of free fatty acids in raw materials significantly reduces the cost of biodiesel production [109]. The use of IR spectroscopy combined with special data treatment procedures (different types of discriminant analysis) makes it possible to choose the cheapest raw material. Like in the case of Raman spectroscopy, NIR spectroscopy provides a possibility to evaluate the transesterification process. For example, authors of the study [110] reported they controlled the esterification reaction by NIR spectroscopy using overtones of C–H and C=O bonds and by multivariate calibration using the principal component analysis with the further discriminant analysis.

#### 4.2.3. Evaluation of the State of Living Organisms

A possibility to obtain information about functional groups of different compounds and to evaluate changes occurring in these compounds under various external influences, which is provided by different types of IR spectroscopy, allows one to use this method for in situ and even in vivo evaluation of the state of living organisms. It is also possible to evaluate changes in the state of the environment by observing changes in living organisms.

For example, a combination of NIR spectroscopy and the PLS-DA mathematical simulator for the study of freely growing leaves of *C. arabica* makes it possible to use these leaves for the monitoring of atmospheric CO_2_ [103].

IR spectroscopy was successfully used for evaluating the biosorption of copper, lead, and cadmium ions by fungal strains, cyanobacteria, and other microorganisms as well as for determining the mechanisms of their adsorption (reviewed in [92]); it was shown that such adsorption occurs on the surface of the above-mentioned microorganisms via the binding of metal ions by compounds containing C=O groups that results in their modification to C-O-X, where X is a metal ion.

Different types of IR spectroscopy are also used to control the processes of bioaccumulation, bioremediation (soil recovery by living organisms), biodegradation (microbial destruction of environmental pollutants), and biomineralization (formation of inorganic solid substances in living organisms) in plants and microorganisms (both single cells and biofilms) [92,111,112,113,114,115].

IR spectroscopy is also used for the rapid identification and classification of microorganisms, though this method is not considered for use as a routine analysis [33,116]. In this case, the experimental design requires the same incubation time and the same composition of a nutrient medium for the studied colonies; the sample should be obtained in the form of a film, and the obtained spectra should be analyzed using the corresponding set of reference spectra required for reliable sample classification. Such approach was successfully used for (*a*) identification of closely-related *Acinetobacter* species (*A. baumannii*, *A. nosocomialis*, *A. pittii*, and *A. calcoaceticus*) with the analytic specificity varied between 90 and 100%; (*b*) distinguishing of *B. cepacia* from other Gram-negative bacteria with the 93.8% specificity; (*c*) identification of *Yersinia* species (*Y. enterocolitica*, *Y. pseudotuberculosis*, *Y. bercovieri*, *Y. intermedia*, etc.) with up to 78.7% specificity; (*d*) identification of Gram-negative clones of some bacteria (*Pseudomonas aeruginosa, Klebsiella pneumoniae, Enterobacter cloacae*, and *Acinetobacter baumannii*) with the exact identification of *P. aeruginosa*, *K. pneumoniae*, and *E. cloacae* strains belonging to the same sequence type; etc. (reviewed in [33]).

#### 4.2.4. The Use of IR Spectroscopy in Agriculture

IR spectroscopy (especially NIR spectroscopy) can be used in agriculture to analyze the composition of initial raw material, to control the quality of agricultural products (including milk) and the process of their production, and to evaluate the composition of feces and manure (reviewed in [92,118,121]); these applications may also be realized in the online mode. A variety of equipment based on IR spectroscopy is also used to control the quality of grain and food rations (including their composition, humidity, homogeneity, etc.) (see, e.g., [117,118,119]).

IR spectroscopy is also used to control the feces of ruminants for the assessment of the digestion quality in animals in terms of food digestibility as well as for the analysis of the stool composition (for example, the starch content). Another important area of IR spectroscopy use in agriculture is the analysis of the chemical composition of slurry and manure [120]. Such an analysis is required for the evaluation of the content of beneficial compounds (total nitrogen, ammonia nitrogen, phosphorus, and potassium) and the quality of the material (moisture content, total carbon content, etc.) used for soil fertilization or biofuel production; it can also be required to control environmental contamination since livestock farms generate ammonia and greenhouse gases. IR spectroscopy is also used for the control of atmospheric emissions of ammonia, nitric oxide, methane, and other volatile organic compounds.

The NIR and mid-IR-based equipment is used to control the milk state in milking machines. In recent years, NIR-based equipment has become more preferable since such devices are relatively cheap compared to mid-IR-based ones. Use of IR spectroscopy makes it possible to evaluate the content of proteins, lipids, urea, lactose, ketone bodies, and somatic cells in milk (a high content of the latter ones may indicate mastitis and poor milk quality) [118].

#### 4.2.5. Studies with the Use of Raman and IR Spectroscopy

In studies based on the use of Raman spectroscopy, the acquisition of IR spectra often represents an additional research method. For example, both IR and Raman spectroscopy were used to confirm the presence of aragonite in shells [73], the presence and conformation changes of carotenoids in algal cells [48], or the structural characterization of epicuticular waxes, sophisticated chemical mixtures of long-chain fatty acids, and their derivatives excreted by plants [102].

## 5. Conclusions

The above-mentioned examples allow us to assert the efficiency of various types of IR and Raman spectroscopy as the tools for environmental measurements. Among different aspects of such use of these approaches, the most promising are: evaluation of the environment contamination with heavy metals by their presence in microbial cultures; indication of the presence of algae or other microorganisms in analyzed samples by the products of their vital activity (that is especially efficient in the case of carotenoids or lipids); evaluation of the oxygen content in the atmosphere by the state of plant leaves; evaluation of the state of wood to determine its readiness to cutting, the influence of environmental factors, or potential fire hazard; evaluation of feces, manure, and other waste products of cattle or wild animals to determine their state, etc.

Though the vibration spectra obtained by these methods are determined by the vibrations of molecular bonds, the results of such measurements can be interpreted as the more “simple” concepts expressing the obtained result either quantitatively (concentration, number of units, etc.) and/or evaluating the result “in terms of probability” (“the compound concentration is… with a 95% probability”, “the sample contains pollutants, which concentration exceeds... with a 80% probability”, “this microbial culture belongs to the genus... with a...probability”, “this sample is eatable with a 99% probability”, “this animal is in a good condition with a...probability”). Such transformation of measurement results became possible due to a well-developed mathematical apparatus, especially in the case of IR spectroscopy, and significantly facilitates the use of such approaches in field studies.

Compared to analytical methods (e.g., such as chromatography), the main drawback of various types of IR and Raman spectroscopy is an insufficient analytical specificity in the case of a sample consisting of several components; however, the authors of such studies consider this drawback to be completely compensated by the noninvasiveness of the method and the rapidity with which results can be obtained.

The main obstacles to the use of various types of oscillation spectroscopy for environmental studies are the complex sample preparation procedure; problems with the measurements in aqueous solutions (in the case of mid-IR spectroscopy); a lack of compact and cheap equipment (especially for Raman spectroscopy); and low sensitivity (especially for NIR spectroscopy). Roughly speaking, such equipment is too expensive and bulky for field studies but is insufficiently precise for laboratory studies. Nevertheless, these drawbacks can be overcome. In the case of in situ or even field studies, the results of a measurement are not always required instantly; sometimes it is enough just to collect samples, which removes the main obstacle to the use of oscillation spectroscopy for environmental studies. Moreover, the existing methods for the post-treatment of obtained spectra partially compensate for the low sensitivity of the method and some requirements for the sample preparation for IR spectroscopy; meanwhile, the evolution of the instrument-making industry resulted in the appearance of inexpensive and compact spectrometers.

NIR spectroscopy seems to be the most promising technology for use in environmental studies and biomonitoring because it is less exacting on the quality of preparations, is able to perform measurements even in the aqueous medium (compared to mid-IR spectroscopy), requires less expensive equipment, and has a well-developed mathematical apparatus (compared to Raman spectroscopy). In addition, as we have already mentioned, devices based on NIR spectroscopy are used (including the distant mode) to monitor the state of animals and milk on husbandry farms, to control the state of wood, etc., while other approaches are less actively used for these purposes.

Thus, various types of IR and Raman spectroscopy can be used or are already used for environmental studies and seem to be even more actively used in the future (especially the NIR spectroscopy).

## Figures and Tables

**Figure 1 ijms-24-06947-f001:**
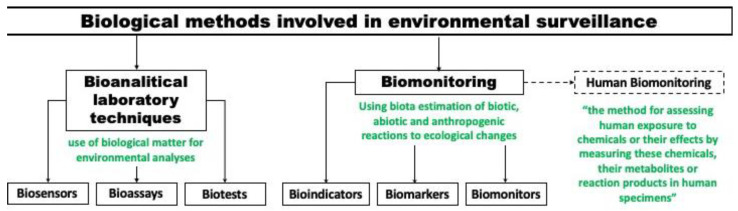
Biological methods of environmental studies.

**Figure 2 ijms-24-06947-f002:**
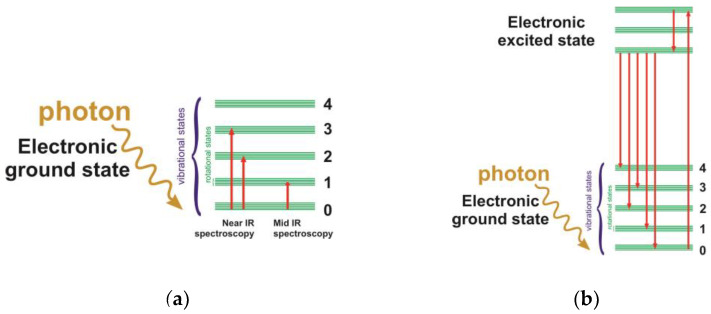
Energy transitions typical for the measurements by infrared and Raman spectroscopy are: (**a**) IR absorption (energy transitions corresponding to the overtone or fundamental transition are shown for near-IR and mid-IR spectroscopy, respectively); (**b**) photoluminescence, which often occurs during the study of biological objects by Raman spectroscopy; and (**c**) various Raman radiation types. Adapted from [29].

**Figure 3 ijms-24-06947-f003:**
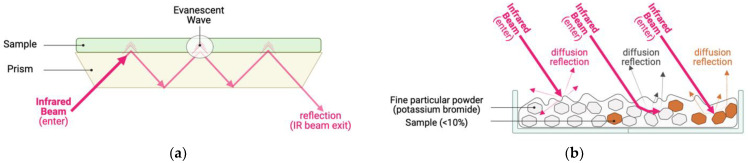
Schematic diagram of various measurement modes: (**a**) Fourier transform infrared spectroscopy with attenuated total reflection (ATR-FTIRS); (**b**) Diffuse Reflectance Infrared Fourier Transform Spectroscopy (DRIFTS).

**Figure 4 ijms-24-06947-f004:**
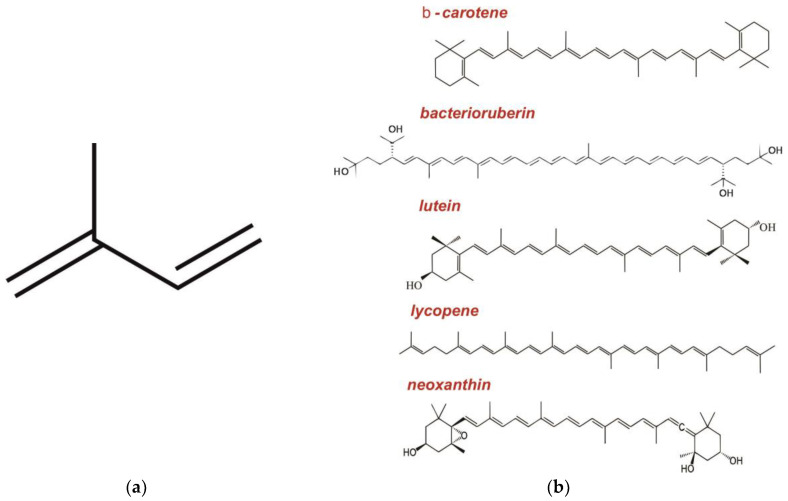
Structures and spectra of different carotenoids: (**a**) Isoprene unit; (**b**) Structures of different carotenoids. Adapted from [78].

**Figure 5 ijms-24-06947-f005:**
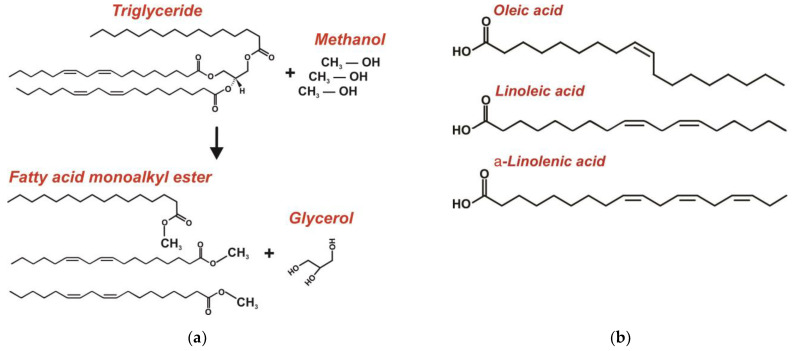
Biofuel sources and production: (**a**) Esterification reaction; (**b**) Some types of fatty acids. Adapted from [82].

**Table 1 ijms-24-06947-t001:** Spectral ranges used in molecular spectroscopy [5].

Region	λ, nm	$\overset{-}{\nu }$, cm^−1^
UV	250–400	40,000–25,000
Visible	400–800	25,000–13,000
Near-infrared	800–2500	13,000–4000
Mid-infrared	2500–25,000	4000–400
Far-infrared	25,000–100,000	400–100

**Table 2 ijms-24-06947-t002:** Peculiar properties of vibrational spectroscopy.

	Near-IR	Mid-IR	Raman
Type of spectrum	Vibrational	Vibrational	Vibrational
Spectral region, cm^−1^	13,000–4000	4000–400	4000–250
Mechanism	Absorption	Absorption	Inelastic scattering
Method sensitive to	Change of molecular dipole moment (polar moieties), associated with overtones and combination bands	Changes of molecular dipole moment (polar moieties), associated mainly with vibrations of various functional groups of molecules	Changes in the molecular polarizability (nonpolar moieties)
Measurement techniques	Fourier Transform Infrared (FTIR) spectroscopy	Fourier Transform Infrared (FTIR) spectroscopy	Resonance Raman spectroscopy
			Surface-enhanced Raman spectroscopy (SERS)
	Transmission spectroscopy	Transmission spectroscopy	Stimulated Raman scattering (SRS)
			Coherent anti-Stokes Raman spectroscopy (CARS)
	Diffuse reflection techniques Fourier Transform Infrared (FTIR) spectroscopy	Diffuse reflection techniques	Tip-enhanced Raman spectroscopy (TERS)
		Attenuated Total Reflectance (ATR	Hyper Raman spectroscopy (HRS)
Microspectroscopy (Microscopy)	Available	Available	Available
Problems	Insufficient analytical specificity in the case of a sample consisting of several components	Insufficient analytical specificity in the case of a sample consisting of several components	Insufficient analytical specificity in the case of a sample consisting of several components
		Limitations in the case of dealing with aqueous solutions (strong absorption of water): interfering signals from atmospheric water and CO_2_, need to dry the samples	Very low signal intensity of the Raman spectrum bands that requires enhanced laser power and a well-developed mathematical apparatus for treatment of spectra
	Low sensitivity (compared to the mid-IR)	Strong absorption of glass adsorption leads to limited use of glass optics and accessories	The photoluminescence spectrum of a sample may overlap with the Raman spectrum, especially in pigment-containing cells and samples, since this significantly complicates the recognition of lines and the interpretation of spectra
			Laser heating and thermodestruction of the sample
	Overlapping substance bands in the NIR spectra and, as a result, difficult spectral interpretation, requiring well-developed mathematical apparatus for treatment of spectra	Low resolution (compared to the Raman) in microscopic studies	Rapid damage to radiated samples requires very careful selection of the radiation power and duration for each biological object
			Rather expensive
Advantages	Noninvasiveness, non-damaging or slightly damaging effects (even compared to the Raman spectroscopy)	Noninvasiveness, non-damaging or slightly damaging effects (even compared to the Raman spectroscopy)	Noninvasiveness, slightly damaging
	Rapid obtaining of results	Rapid obtaining of results	Rapid obtaining of results
	Cheap method compared to its analogues	Cheap method compared to its analogues	-
	Minimal sample preparation	-	Minimal sample preparation
	Possibility to evaluate polar molecules (alcohols, phenolic compounds, etc.)	Possibility to evaluate polar molecules (alcohols, phenolic compounds, etc.)	Measuring in water solutions, suitability of glass optics, and accessories
	-	Good resolution and sensitivity compared to NIR	High resolution (compared to IR techniques)
Suitability for using as the tools for environmental measurements	Yes	Yes	Yes

**Table 3 ijms-24-06947-t003:** Substances in cells observed using Raman spectroscopy.

Substance	Objects	Methods	Bands, cm^−1^	Used for Estimation	Ref.
Carotenoids and other polyenes (e.g., β-carotene, lutein, lycopene, astaxanthin)	Microalgae, extremophiles, lichens; Watermelon, hot pepper, skin	Resonance Raman spectroscopy, Raman microscopy; SERS	~1530, ~1120–1200, and ~1000	Carotenoids detection; Cell detection; Stress detection Concentration of carotenoids; Maturity degree	[42,43,44,45,46,47,48,49,50,51,52,53,54,55]
Chlorophyll	Microorganisms (algae, bacteria, lichens, and endoliths)	Raman spectroscopy, Resonance Raman spectroscopy, Raman microscopy	Wide range e.g., ~1640, ~1554, ~1437, ~1325, ~1289, ~1233, ~1186, ~1173, ~1020, ~986, ~915	Organisms’ condition	Reviewed in [43,56]
Quinones, flavonoids (polyphenols), pheophytin, cytochromes, hemoglobin et al.; Flexirubins, scytonemin (specific for cyanobacteria)	Microorganisms (algae, bacteria, lichens, and endoliths)	Raman spectroscopy, Resonance Raman spectroscopy, Raman microscopy	Wide range	Organisms’ condition Cell detection;	Reviewed in [43,57,58]
Triglycerides (oil) and fatty acids	Algae and microalgae	Raman spectroscopy, Resonance Raman spectroscopy, Raman microscopy; SERS	1650 (unsaturated) and 1440 (saturated) bonds, respectively	Biofuel production; Oil concentration	[56,59,60,61,62,63]
Ethanol	Lignin and cellulose conversion to bioethanol in bioreactor	Raman spectroscopy	883	Biofuel production	[59,64]
Glucose and Glucose oligomers and polysaccharides	Lignin and cellulose conversion to bioethanol in bioreactor	Raman spectroscopy	1120 Typical glycosidic bond peak at 920–960	Biofuel production	[59]
Inorganic polyphosphate inclusions	Microalgae	Raman microscopy	Peak at ~1160	Substance and concentration detection	[65,66]
Nitrogen (crystalline guanine)	Microalgae	Raman microscopy	Maximum at 651	Substance and concentration detection	[67,68]
Starch	Microalgae	Raman microscopy	Maximum ~479	Substance and concentration detection	[68]
Lipids	Microalgae	Raman microscopy	Maximum ~2854	Substance and concentration detection	[68]
Calcite and aragonite magnesite	Shells (exoskeletons)	Raman spectroscopy, Raman microscopy	Peaks at 156, 282, and 1087 (calcite); Peaks at 155, 206, and 1086 (aragonite); Band at 1089 (magnesite)	Assessment of the size and mineralization level	[69,70,71,72,73,74]
Microorganisms, strains of micro-organisms	Specific binding cell sites	SERS with special particles containing biorecognition molecules	The peak of particles	Ultrasensitive bacteria (pathogen) detection	[75,76]

**Table 4 ijms-24-06947-t004:** Substances in cells observed using IR spectroscopy.

Objects	Substances	Methods *	Used for Estimation	Ref.
Plants, fruits	Antioxidants (e.g., phenolic compounds, carotenoids, triterpenoids); organic acids; polysaccharides; essential oils; fatty acids	NIR and IR spectroscopy	Qualitative and quantitative detection	[94,95,96,97,98,99,100,101,102]
Leaves of *C. arabica*	Compounds of leaves	NIR spectroscopy	Monitoring of atmospheric CO_2_	[103]
Wood	Cellulose, lignine, phenolic compounds, and other wood components	Mainly NIR, DRIFT-MIR	Evaluation chemical structure and physical properties (including in vivo); changes in composition; density; degradation level; humidity	[104,105,106,107,108]
Control for: Raw materials; transesterification process in bioreactor	Oils, fats, free fatty acids; methyl esters;	NIR and IR spectroscopy	Biodiesel quality control: choice of the raw materials; reduction of cost; control of trans-esterification reaction; determination of the properties, quality, and contaminants of biodiesel	[61,109,110]
Fungal strains, cyanobacteria, other microorganisms and plants; single cells and biofilms	Compounds containing C=O groups	IR spectroscopy	Evaluating biosorption of copper, lead, and cadmium ions; control of bioremediation, biodegradation, and biomineralization	[92,111,112,113,114,115]
Microorganisms, strains of microorganisms	Compounds (molecules) containing in microorganisms	IR spectroscopy	Rapid identification and classification of microorganisms	Reviewed in [33,116]
Agriculture: faeces and manure; agricultural products (including grains and milk)	Compounds (molecules) containing in studied substances	NIR (mainly) and IR spectroscopy	Control the quality of agricultural products; control the quality of grain and food rations (including their composition, humidity, homogeneity, etc.); evaluate the composition of faeces and manure for the assessment of the digestion quality in animals and analysis of a stool composition; control of atmospheric emissions of ammonia, nitric oxide, methane, and other volatile organic compounds; control the environmental contamination (ammonia and greenhouse gases)	[92,117,118,119,120,121]

* Also include the special mathematical apparatus for the interpretation of the results of measurement.

## Data Availability

Data available in a publicly accessible repository and is listed in the list of references.

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
