# Peer review of "Vibrational Spectroscopy as a Tool for Bioanalytical and Biomonitoring Studies"

_ijms, 2023, doi:10.3390/ijms24086947_

Round 1

Reviewer 1 Report

Within the framework of the review, a synthesis of the relevant and current literature for the approached field was made.

The study is fully justified and motivated, as it fills the existing gaps in terms of a synthetic presentation of analysis methods related to vibrational spectroscopy in the study of the environment.

The introduction is well designed, appropriately reflecting the current state of the art in the field of review addressed.

The presented spectroscopy methods in order to carry out the study are presented clear and in a well-structured manner.

Most of the references cited are more recent than 5 years.

The contribution and added value of the study is supported by the following aspects:

- Different types of IR and Raman spectroscopy can be used as effective tools for measuring environmental quality;

- The results of vibrational spectral measurements can be interpreted and expressed both quantitatively and/or in terms of probability;

- The various types of IR and Raman spectroscopy are non-invasive and allow rapid results to be obtained;

- NIR spectroscopy can be considered the most suitable technology in environmental and biomonitoring studies because it is less demanding, can perform measurements even in aqueous environments, requires less expensive equipment and has a well-developed mathematical apparatus;

- Different types of IR and Raman spectroscopy, especially NIR spectroscopy, will be increasingly used in future environmental studies.

Author Response

Response to Reviewer 1 Comments

Article: ” Potentialities of the application of a vibrational spectroscopy in environmental studies”

Authors: Sergey K. Pirutin, Shunchao Jia, Alexander I. Yusipovich, Mikhail A. Shank*, Evgeniia Yu. Parshina and Andrey B. Rubin

Within the framework of the review, a synthesis of the relevant and current literature for the approached field was made.

The study is fully justified and motivated, as it fills the existing gaps in terms of a synthetic presentation of analysis methods related to vibrational spectroscopy in the study of the environment.

The introduction is well designed, appropriately reflecting the current state of the art in the field of review addressed.

The presented spectroscopy methods in order to carry out the study are presented clear and in a well-structured manner.

Most of the references cited are more recent than 5 years.

The contribution and added value of the study is supported by the following aspects:

-           Different types of IR and Raman spectroscopy can be used as effective tools for measuring environmental quality;

-           The results of vibrational spectral measurements can be interpreted and expressed both quantitatively and/or in terms of probability;

-           The various types of IR and Raman spectroscopy are non-invasive and allow rapid results to be obtained;

-           NIR spectroscopy can be considered the most suitable technology in environmental and biomonitoring studies because it is less demanding, can perform measurements even in aqueous environments, requires less expensive equipment and has a well-developed mathematical apparatus;

-           Different types of IR and Raman spectroscopy, especially NIR spectroscopy, will be increasingly used in future environmental studies.

Good Day!

Dear Reviewer,

Thank you for your positive and thorough review of our article. We are pleased to hear that you found our manuscript to be well-justified and motivated, and that it fills existing gaps in the presentation of analysis methods related to vibrational spectroscopy in environmental studies.

We are glad that you found our introduction to be well-designed and reflective of the current state of the art in the field of review addressed. We are also happy to hear that you found the presentation of the spectroscopy methods to be clear and well-structured.

We appreciate your feedback on the references cited in our article and aspects about value of the study..

We would like to highlight that in our manuscript, we have added, summarized and systematized the information provided into three tables: Table 2. Substances in cells observed using Raman spectroscopy, Table 3. Substances in cells observed using IR spectroscopy, and Table 4. Peculiar properties of Near-, Mid-IR, and Raman spectroscopy. These tables provide a clear and concise overview of the information presented in the review.

This adding of tables has enabled us to increase the number of sources in the reference list to 115.

Lastly, we would like to acknowledge the correction made to the horizontal axis name of Raman spectroscopy, which has been updated to "Raman shift".

Thank you again for taking the time to review our article, and for your insightful comments.

Thank You!

Best regards,

Authors: Sergey K. Pirutin, Shunchao Jia, Alexander I. Yusipovich, Mikhail A. Shank*, Evgeniia Yu. Parshina and Andrey B. Rubin

Reviewer 2 Report

In the manuscript, the author represents: “Potentialities of the application of a vibrational spectroscopy in environmental studies”. The manuscript is too short for the article review. In my opinion, this is the book chapter more than the article review due to the without discussion and summary. Moreover, the author needs to clearly answer some issues:

1.    The “basic principles” are too long (8/20 pages from number 4 to 12). The author needs to summarize the main points.

2.    With IR spectroscopy method, the spectra are largely absorbed and influenced by the presence of water and CO2. What happens if you use IR in the water or air environmental studies

3.    The number of papers in the manuscript is too little and not systematic. The author needs to summarize by the tables.

4.    The horizontal axis name of Raman spectroscopy should be “Raman shift”.

Author Response

Response to Reviewer 2 Comments

In the manuscript, the author represents: “Potentialities of the application of a vibrational spectroscopy in environmental studies”. The manuscript is too short for the article review. In my opinion, this is the book chapter more than the article review due to the without discussion and summary. Moreover, the author needs to clearly answer some issues:

Good Day!

Thank you for the review. Indeed, the manuscript text resembles a chapter from a book, especially in terms of its length. However, it was intended to be a mini-review format, which does not require a large amount of text, especially since various types of vibrational spectroscopy are not very actively used for environmental research at the moment (excluding human monitoring).

Point 1. The “basic principles” are too long (8/20 pages from number 4 to 12). The author needs to summarize the main points.

Response 1: The paper is written at the intersection of several scientific disciplines, and is intended to be read by a relatively wide audience, describing various methods and concepts of vibrational spectroscopy. However, unfortunately, when describing the basics of instrument operation, measurement procedures, or measurement results used in vibrational spectroscopy, terminology and concepts that are not well understood by field researchers are traditionally used, even though it is not always necessary. Therefore, we were forced to include a fairly large section in the paper dedicated to theory and method description, which, as we hope, is presented in a relatively simple and understandable language. We tried to shorten this section, but it was difficult to do so due to the specificity and relative obscurity of vibrational spectroscopy methods. We also summarize the main points of this section in Table 4 (984 line).

Point 2: With IR spectroscopy method, the spectra are largely absorbed and influenced by the presence of water and CO2. What happens if you use IR in the water or air environmental studies

Response 2: Yes, it has been corrected. Some of this information was provided in the relevant sections of the text dedicated to mid and near-infrared spectroscopy. Additionally, a new section titled "The Effect of Water and Atmospheric Carbon Dioxide on IR Spectra" was added in section 3.3 Infrared Spectroscopy: “The effect of water and atmospheric carbon dioxide on IR spectra. IR spectra (especially mid-IR spectra) are largely absorbed and influenced by the presence of water and CO2 (in case of air measurements). Water, as dipole molecule, actively absorbs IR radiation, and have an IR spectrum with strong broad -OH peak in the 3500-3000 cm-1 range and a less wide -OH peak in the region ~1650 cm-1 (in addition, there are several less intense peaks in the spectrum of water, e.g. 2100, 710-645 cm-1, as well as 3250 cm-1 overtones) that will cover up a lot of other substances peaks, such as amides, lipids, proteins, alcohols. This makes it difficult to identify and interpret spectra (furthermore, water is not a very good solvent for IR samples). This is not really matter in case of NIR spectra, the overtones of –OH groups do not make such a significant contribution, as with mid-IR spectroscopy. In the case of mid-IR spectroscopy, special sample preparation is used to minimize the effect of the water signal; in addition, the using infrared spectroscopy with attenuated total reflection allows recording a signal in a thin layer of a substance, which also reduces the absorption of water. As a rule, when registering a signal in aqueous solutions, the spectrum of water is subtracted from the spectrum of the sample. In the case of biological experiments, the influence of CO2 molecules, unlike water, is not as significant due to its low concentration in the air. In addition, the most intense band of the IR spectrum of CO2 (~2300-2400 cm-1) does not overlap with the bands of the spectra of most biological molecules”

Corresponding information was also included in Table 4: ”… limitations in the case of dealing with aqueous solutions (strong absorption of water): interfering signals from atmospheric water and CO2, need to dry the samples;”.

Point 3. The number of papers in the manuscript is too little and not systematic. The author needs to summarize by the tables.

Response 3: Yes, we summarized and systematized the provided information in manuscript into tables (Table 2. Substances in cells observed using Raman spectroscopy, Table 3. Substances in cells observed using IR spectroscopy and Table 4. Peculiar properties of Near-, Mid-IR and Raman spectroscopy.).

It is worth noting that the topic of our scientific review is quite new when it comes to environmental research without human biomonitoring. Additionally, we refer to a significant amount of scientific reviews, and it is worth mentioning that there is not a lot of literature on this topic. Nonetheless, we have included three summarized tables in our review article, and thus increased the number of sources in the reference list to 115.

Point 4. The horizontal axis name of Raman spectroscopy should be “Raman shift”.

Response 4: The horizontal axis name of Raman spectroscopy was corrected to “Raman shift”

Thank You!

Best regards,

Authors: Sergey K. Pirutin, Shunchao Jia, Alexander I. Yusipovich, Mikhail A. Shank*, Evgeniia Yu. Parshina and Andrey B. Rubin

Round 2

Reviewer 2 Report

1. The author needs to delete all comments in the manuscript.

2. The author needs to double-check the font, color letter in manuscript.

3. The author don't summarized and change the "basic principle" on my opinion.

Author Response

Dear Reviewer,

Thank you for taking the time to review our manuscript:

Point 1. The author needs to delete all comments in the manuscript.

Response 1. Following your suggestion, we have deleted all comments from the manuscript.

Point 2. The author needs to double-check the font, color letter in manuscript.

Response 2. After carefully reviewing the manuscript, we have double-checked the font and color of all letters in the manuscript. We have ensured that the font and color are consistent throughout the manuscript, and that they are easily readable.

Point 3. The author don't summarized and change the "basic principle" on my opinion.

Response 3. After carefully considering your comments, we have made several changes to the manuscript to improve its readability and orientation. We have made changes in the third section (Vibrational spectroscopy and its types. A brief theory): slightly reduced, restructured text, adding subsections name to systematized the content and finalizing and summerizing the data in Table 2. We have also included the last paragraph mentioning Table 2. Furthermore, we have made changes to the content of Table 2 by adding two new items. We have rearranged the tables, with the former fourth table now being moved to the third section and renumbered as Table 2 (which mantioned above), this table summarizes the information from section three. The second table from previous variant of the manuscript has become the third now, and the third table has become the fourth now. We hope that this will help to systematize and summarize the information contained in this section.

As we wrote earlier, the paper is written at the intersection of several scientific disciplines, and is intended to be read by a relatively wide audience, describing various methods and concepts of vibrational spectroscopy. However, unfortunately, when describing the basics of instrument operation, measurement procedures, or measurement results used in vibrational spectroscopy, terminology and concepts that are not well understood by field researchers are traditionally used, even though it is not always necessary. Therefore, we were forced to include a fairly large section in the paper dedicated to theory and method description, which, as we hope, is presented in a relatively simple and understandable language. We tried to shorten this section, but it was difficult to do so due to the specificity and relative obscurity of vibrational spectroscopy methods.

We understand the importance of presenting our research findings in a concise and clear manner. However, in this particular section, we need to provide detailed explanations and descriptions to ensure that the reader fully grasps the underlying concepts and methodologies. Any attempts to reduce the length of the section would compromise the clarity and accuracy of our results, which could ultimately lead to misunderstandings.

We have taken your feedback into account and have re-reviewed the section to ensure that it is as clear and concise as possible. We believe that our revisions have improved the clarity of the section while maintaining its significance and relevance to the study.

We believe that these changes have addressed the issues raised in your comment and improved the quality of the manuscript.

Thank you again for your valuable feedback, which has helped us to improve our work.

Thank You!

Best regards,

Authors: Sergey K. Pirutin, Shunchao Jia, Alexander I. Yusipovich, Mikhail A. Shank*, Evgeniia Yu. Parshina and Andrey B. Rubin

Round 3

Reviewer 2 Report

1. The author needs to double-check equation (5).

2. The author don't know how to summary the basic principle.

Ex: equation (1) can combine with equation (2).

3. Format of Fig. 3 is mistake. The author needs to double-check it.

4. The author needs to format the layout of all tables. The author needs to explore more the references.

5. The title doesn't fix with content of manuscript. In the title, author want to focus in "environmental studies", but the content is "biological studies". The author needs to change the title.

Author Response

Point 1. The author needs to double-check equation (5)

Response 1. Thank you for the information regarding equation (5). We have thoroughly reviewed it and agree with your observation that the subscript format was enlarged. We have rectified this and returned it to the original subscript format. We appreciate your attention to detail and assistance in improving our work.

Point 2. The author don't know how to summary the basic principle.

Ex: equation (1) can combine with equation (2).

Response 2. Unfortunately, we do not fully understand this recommendation. In our opinion, the example given to combine equation 1 and 2 only highlights the difficulty and impossibility of summarizing and reducing the text. Equation 1 describes the relationship between wavelength and frequency, while formula 2 describes the relationship between energy and wavelength, and is related to energy quantization. The value of the symbol "nu bar" (written as a horizontal line over the symbol "nu") is similar to "nu" in meaning, but it represents an energy quantity, and "frequency" is a conditional term. Nonetheless, we appreciate your attention to this section of the text. We acknowledge that these two formulas cannot be in the same subsection, and have therefore divided section 3.1.1 into two parts. Of course, we could remove formula 1 and partially remove the accompanying text. In this case (this is not always possible in other cases), this would not greatly affect the contents of the manuscript. However, only the formula and 1-2 sentences can be painlessly reduced (the remaining ones are needed to describe the variables in formula (2). Thus, the gain in the amount of text will be insignificant, but some information will be lost. It seems reducing the size in this case is not worth it. Especially given the need, as was already mentioned, to explain the basics of instrument operation, measurement procedures, or measurement results used in vibrational spectroscopy, terminology and concepts by field researchers as potential readers of this article.

Point 3. Format of Fig. 3 is mistake. The author needs to double-check it.

Response 3. Yes, thank You, we changed the Figure arrangement format from "stacked" to "side-by-side (left and right)" in accordance with the IJMS template.

Point 4. The author needs to format the layout of all tables. The author needs to explore more the references

Response 4. Yes, thank You, we made the necessary changes to the work. All tables were formatted in accordance with the IJMS template requirements. In addition, we added references to the table and text of manuscript.

Point 5. The title doesn't fix with content of manuscript. In the title, author want to focus in "environmental studies", but the content is "biological studies". The author needs to change the title.

Response 5. Yes, we agree, title doesn't fix with content of manuscript, we change the title for “Vibrational spectroscopy as a tool for bioanalytical and biomonitoring studies”
